# Antibacterial Activities of Crude Secondary Metabolite Extracts from *Pantoea* Species Obtained from the Stem of *Solanum mauritianum* and Their Effects on Two Cancer Cell Lines

**DOI:** 10.3390/ijerph16040602

**Published:** 2019-02-19

**Authors:** Nkemdinma Uche-Okereafor, Tendani Sebola, Kudzanai Tapfuma, Lukhanyo Mekuto, Ezekiel Green, Vuyo Mavumengwana

**Affiliations:** 1Department of Biotechnology and Food Technology, Faculty of Science, University of Johannesburg, PO Box 17011, Doornfontein, Johannesburg 2028, South Africa; tendanisebola@rocketmail.com; (T.S.); kudzanai2003@gmail.com (K.T.); egreen@uj.ac.za (E.G.); 2Department of Chemical Engineering, Faculty of Engineering and the Built Environment, University of Johannesburg, PO Box 17011, Doornfontein, Johannesburg 2028, South Africa; lukhanyom@uj.ac.za; 3South African Medical Research Council Centre for Tuberculosis Research, Division of Molecular Biology and Human Genetics, Department of Medicine and Health Sciences, Stellenbosch University, Tygerberg 7505, South Africa

**Keywords:** antibacterial, anticancer, bacterial endophytes, MIC plants, *Solanum mauritianum*, *16S* rRNA

## Abstract

Endophytes are microorganisms that are perceived as non-pathogenic symbionts found inside plants since they cause no symptoms of disease on the host plant. Soil conditions and geography among other factors contribute to the type(s) of endophytes isolated from plants. Our research interest is the antibacterial activity of secondary metabolite crude extracts from the medicinal plant *Solanum mauritianum* and its bacterial endophytes. Fresh, healthy stems of *S. mauritianum* were collected, washed, surface sterilized, macerated in PBS, inoculated in the nutrient agar plates, and incubated for 5 days at 30 °C. Amplification and sequencing of the *16S* rRNA gene was applied to identify the isolated bacterial endophytes. These endophytes were then grown in nutrient broth for 7–14 days, after which sterilized Amberlite^®^ XAD7HP 20–60 mesh (Merck KGaA, Darmstadt, Germany) resin was added to each culture to adsorb the secondary metabolites, which were later extracted using ethyl acetate. Concentrated crude extracts from each bacterial endophyte were tested for antibacterial activity against 11 pathogenic bacteria and two human cancer cell lines. In this study, a total of three bacterial endophytes of the *Pantoea* genus were identified from the stem of *S. mauritianum*. The antibacterial test showed that crude secondary metabolites of the endophytes and stem of *S. mauritianum* possessed antibacterial properties against pathogenic microbes such as *Escherichia coli, Staphylococcus aureus, Klebsiella pneumoniae,* and *Pseudomonas aeruginosa*, with concentrations showing inhibition ranging from 0.0625 to 8.0000 mg/mL. The anticancer analysis showed an increase in cell proliferation when A549 lung carcinoma and UMG87 glioblastoma cell lines were treated with both the plant and endophytes’ crude extracts. As far as we know, this is the first study of its kind on *Solanum mauritianum* in South Africa showing *S. mauritianum* endophytes having activity against some of the common human pathogenic organisms.

## 1. Introduction

Infectious diseases caused by bacteria, fungi, viruses, and parasites are major public health concerns, despite the remarkable progress in human medicine. Their impact is exceptionally great in developing countries due to the relative unavailability of effective medicines and the rise of widespread drug resistance [1,2]. The emergence and spread of antimicrobial resistant pathogens is progressively increasing, making the current available antimicrobials less effective and in need of reviewing. In recent years, there has been a growing awareness and interest in the study of medicinal plants and their endophytes as alternative sources for bioactive compounds against resistant microorganisms [3,4].

Endophytes are microorganisms that live or spend a part of their life cycle in the internal tissues of the plants without causing any symptoms of disease in the plant [5,6,7,8,9,10,11,12]. In certain instances, endophytes can mimic the chemistry of their respective host plants to produce similar bioactive natural products, or derivatives that are more bioactive than those of their hosts [13,14]. An example can be seen with the case of taxol being produced by a series of endophytes from yews as well as other plant sources [15,16]. Endophytes which produce the same important bioactive compounds such as peptides, steroids, phenolic compounds, aliphatic compounds, terpenoids, alkaloids, lignans, and isocoumarins [12], as their host plants not only reduce the need to harvest slow growing and possibly rare plants, but also preserve the world’s ever diminishing biodiversity. Additionally, it is recognized that a microbial source of a valued product may be easier and more economical to produce, effectively reducing its market price [3]. Kakadumycin A, isolated from a *Streptomyces* species found in the plant *Grevillea pteridofolia*, Camptothecin from *Paenibacillus polymyxa* found in *Camptotheca acuminate*, and Asparaginase from *Pseudomonas oryzihabitans*, isolated from *Hibiscus rosasinensis*, are some bioactive compounds extracted from endophytes [12].

Our interest in this study was on the medicinal plant *Solanum mauritianum*, its bacterial endophytes, and the antibacterial activities of secondary metabolite crude extracts of the two. As far as we know, *S. mauritianum* is a plant which has been underexplored with regards to its endophytes. *S. mauritianum* is an invasive weedy plant species from the family Solanaceae, which has been reported to be useful in South African traditional medicine for the treatment of menorrhagia [17,18], dysentery, diarrhoea [18,19], and infertility [18,20]. The main aim of this study was to determine the bacterial endophyte diversity of *S. mauritianum* and the antibacterial activity of both the plant and endophytes’ secondary metabolite crude extracts on selected pathogenic bacteria.

## 2. Materials and Methods

### 2.1. Sample Collection

Fresh, healthy (showing no apparent symptom of disease) stems of *S. mauritinum* plant were collected from the University of Johannesburg, Doornfontein Campus, located in Johannesburg, South Africa. The samples were transported to the laboratory where the plant was thoroughly washed with sterile distilled water and used within four hours.

### 2.2. Endophytes Isolation and Identification

The stems were surface-sterilized separately, using the method described by [12,21]. Briefly, each sample (approximately 10 g) was treated with 5% Tween 20 (enough to cover the plant material) and shaken vigorously for five minutes. The Tween 20 was removed by rinsing several times with sterile distilled water. The sample was then disinfected with 70% ethanol for one minute. Traces of the ethanol were removed by rinsing with sterile distilled water five times. The sample was then treated with 1% Sodium Hypochlorite (NaHClO) for ten minutes and rinsed five times with sterile distilled water. The last rinse was used as a control and 100 µL of it was plated on Nutrient Agar (NA). The sample was then macerated in sterilized phosphate-buffered saline (PBS) with the outer surface trimmed out. The macerated sample was serially diluted up to 10^−3^ dilution and each dilution inoculated (using spread plate method) in triplicates on and Nutrient Agar (NA) (for bacteria enumeration). The NA plates were incubated at 30 °C, (IncoTherm, Labotec, Johannesburg, South Africa). The growth was monitored periodically for 5 days during the incubation period. Effectiveness of the sterilization was monitored on the wash control plate, with growth indicating poor sterilization. Under such circumstances, the plates were discarded, and the sterilization repeated.

Distinct colonies were selected and sub-cultured on the appropriate NA to obtain pure isolates. Pure bacterial isolates were preserved in 50% glycerol on a ratio of 1 mL glycerol to 1 mL overnight broth culture and stored at −80 °C.

#### 2.2.1. Morphological Identification of Bacterial Endophytes

Gram stain technique as described by [22] was done to determine cell morphology.

#### 2.2.2. Molecular Identification

##### Genomic DNA Extraction, Polymerase Chain Reaction, and Sequencing

Genomic DNA of each bacterial endophyte isolate was extracted from pure colonies obtained from nutrient agar plates. The DNA was extracted using Bacterial DNA kit (Zymo Research, catalog NO R2014). The extracted DNA was quantified using the NanoDrop ND-2000 UV-Vis spectrophotometer (Thermo Fisher scientific, Waltham, MA, USA). The *16S* rRNA gene of each bacterial isolate was amplified by PCR following [23]. Briefly, the *16S* rRNA gene was amplified using the primers (16S-27F: 5′-AGAGTTTGATCMTGGCTCAG-3′ and 16S-1492R: 5′-CGGTTACCTTGTTACGACTT-3′) with 2x PCR master mix with standard buffer. The PCR products were cleaned with ExoSAP-it™ following manufacturers’ recommendations and sequencing was done at Inqaba Biotechnical Industries (Pty) Ltd., Pretoria, South Africa.

##### Phylogenetic Analysis

The obtained sequences were screened for chimeras using DECIPHER23 and subjected to BLAST analysis at the National Center for Biotechnology Information (NCBI) against the prokaryotic rRNA sequence database (Bacteria and archaea) to identify the closest bacterial species. Bacterial species with 98–100% similarity were selected for phylogenetic analysis. Alignments of nucleotide sequences (isolate and species obtained through BLAST) were performed using MUSCLE with default options. Phylogenetic trees were constructed using the neighbour-joining (NJ) method based on the Tamura–Nei model. A total of 1000 replications were used for the bootstrap test. All branches with a bootstrap value greater than 50% were considered to be significant. The positions containing gaps and missing nucleotide data were eliminated. All evolutionary analyses were conducted in MEGA 7.27. The *16S* rRNA gene sequences of bacterial isolates identified in the study were deposited in GenBank (www.ncbi.nlm.nih.gov/genbank/) with the accession numbers as stated in Table 1. The assigned names of the bacterial isolates were based on the BLAST homology percentages as well as phylogenetic results.

### 2.3. Biological Activity Assay

#### 2.3.1. Isolation of Secondary Metabolites from Endophytes

Secondary metabolites were extracted from the endophytic bacteria using the method described earlier [24] with modifications. The endophytic bacteria isolated from *S. mauritianum* were cultured in three 5 L Schott bottles each containing 3 L of a nutrient broth and shaken at 200 rpm at 27 °C for seven days. After seven days of cultivation, sterilized Amberlite^®^ XAD7HP 20–60 mesh (Merck KGaA, Darmstadt, Germany) resin (60 g/L) (Sigma–Aldrich, Johannesburg, South Africa) was added to adsorb the organic products, and the culture and resin were shaken at 200 rpm for 2 h. The resin was filtered through cheesecloth and washed three times with 250 mL of acetone for each wash. The acetone soluble fraction was concentrated using a rotary evaporator and a dark brown viscous extract was obtained.

The extract was transferred into a measuring cylinder and based on the volume, ethyl acetate was added in a 1:1 ratio (v/v). The mixture was shaken vigorously for 5–10 min, poured into a separating funnel and allowed to separate; this was done until the dark brown viscous liquid obtained after removing the acetone became a light-yellow liquid. The ethyl acetate fraction was removed using a rotary evaporator and the extract was stored in an amber bottle in a cool dry place until analysis was done. The light-yellow liquid was evaporated (to ensure there was no extract lost) leaving behind no reasonable extract. No further analysis was done on this.

#### 2.3.2. Extraction of the *Solanum Mauritianum* Plant Part

The method described by [25,26] with slight adjustments was used for the preparation of the crude extracts of the stems of *S. mauritianum*. The dried plant part was blended into a fine powder with a shop-bought coffee mill and 200 g of plant powder was weighed into a Schott bottle, 2000 mL of a methanol/chloroform (50:50, v/v) solution was added and the Schott bottle left on a platform shaker for three days. The extract suspension was filtered through Whatman No. 1 filter paper and evaporated to dry out the solvent, using a rotatory evaporator with consideration to the boiling points of the extracting solvents. The process was repeated another two times to ensure maximal extraction of compounds. The crude extract was collected in a beaker and placed in a desiccator to dry out completely.

#### 2.3.3. Determination of Anti-Bacterial Activity

The Minimum Inhibitory Concentrations (MIC) method was used in this study to determine the antibacterial activity of crude extracts from the plant and bacterial endophytes. The following bacterial strains were used: *Bacillus cereus* (ATCC10876), *Bacillus subtilis* (ATCC19659), *Enterobacter aerogenes* (ATTC13048), *Escherichia coli* (ATCC10536), *Klebsiella pneumonia* (ATCC10031), *Mycobacterium smegmatis* (ATCC21293), *Mycobacterium marinum* (ATCC927), *Proteus vulgaris* (ATCC 33420), *Pseudomonas aeruginosa* (ATCC10145), *Staphylococcus aureus* (ATCC25923), *Streptococcus epidermidis* (ATCC14990).

MICs were carried out according to the method outlined by [26] and [27]. The bacterial strains were inoculated into Mueller Hinton (MH) broth and allowed to grow overnight in an incubator at 37 °C for 24 to 36 h, depending on the growth rate of each bacteria, and compared to a 0.5 McFarland’s standard. The antibiotic, Streptomycin, was used as the positive control and was prepared by weighing 0.032 mg in 1 mL of sterile distilled water while DMSO was used as a negative control.

The crude secondary metabolite extracts from the identified endophytes were weighed (0.176 g) into empty autoclaved MacCourtney bottles to ensure sterility. The crude extracts were dissolved in DMSO (0.1%) to make a stock solution of 32 mg/mL. Serial dilutions were carried out using the MH broth from 16 mg/mL down to 0.03125 mg/mL. The outer wells of the plate were filled with sterile distilled water. Standardized overnight bacterial cultures (100 µL) were added into each well horizontally and vertically in 5 repeats for each bacterium. In vertical order, 100 µL of the diluted samples were added in the wells from 16 mg/mL down to 0.03125 mg/mL. The plates were covered and incubated overnight at 37 °C. Resazurin sodium salt solution (10 µL of 0.02% (w/v)) was added to the wells and incubated for another two hours. Upon reduction, resazurin changes colour from blue to pink to clear, as oxygen becomes limited within the medium, indicating metabolism. The well with a known concentration showing a slight colour change was used as MIC. The wells were visually inspected for colour changes.

#### 2.3.4. Anticancer Assays

Crude extracts of *S. mauritianum* stems and secondary metabolites derived from bacterial endophytes were tested against two ATCC cancer cell lines: U87MG Glioblastoma and A549 Lung carcinoma cells for anticancer activity. The crude samples were weighed in Eppendorf tubes, 0.1% DMSO was added and sonicated to aid dissolution, and a stock solution of 200 µg/mL was made. Serial dilutions were carried out using growth media from 100 µg/mL to 3.13 µg/mL. An MTS (3-(4,5-dimethylthiazol-2-yl)-5-(3-carboxymethoxy-phenyl)-2-(4-sulfophenyl)-2H-tetrazolium) in vitro cytotoxicity assay was conducted to determine change in cell viability, through a colour change. An MTS compound (yellow) is metabolized by viable cells to form a dark purple-coloured compound, visible through UV Vis spectroscopy at 490 nm. The absorbance is directly proportional to the cell viability. The samples were analyzed in duplicates across three plates (*n* = 6) and the average value reported. The U87MG cells and A549 cells were grown using normal tissue culture techniques and 15% FBS addition. The cells (1 × 105 cells/mL) were incubated in 96 well plates at 37 °C overnight, with the subsequent addition of the supplied compounds, in concentrations of (100 μg/mL, 50.0 μg/mL, 25.0 μg/mL, 12.5 μg/mL, 6.25 μg/mL, 3.125 μg/mL, and 0 μg/mL). The cells were left to incubate for 4 days, after which MTS (5 μL) was added to the cells. The absorbance values were measured at 490 nm after 1 h, 2 h, and 4 h incubation periods, averaged and the viability curves drawn up. Auranofin was used as a positive control [28], based on its excellent activities against non-small cell lung cancer cells [29,30].

## 3. Results

### 3.1. Identification of Bacterial Endophytes

In this study, three bacterial endophytes (from the Enterobacteriaceae family) of the *Pantoea* genus isolated and identified from the stem of *S. mauritianum* are shown in Table 1.

The accession numbers of the isolated endophytes are also shown in Table 1. All isolates had 99% similarities with other strains obtained from GenBank. The results of the effectiveness of the surface sterilization method showed no microbial growth in the control plates, indicating that the isolates were endophytes (Gram negative rods). Relationships between the three *Pantoea* genus endophytic bacteria isolated from the *S. mauritianum* stem and other species of the same families are shown in Figure 1. The phylogenetic analysis showed that the endophytic bacterial isolates aligned with various closely related bacterial species.

### 3.2. Antibacterial Activity of Bacterial Endophytes and Plant Part

Minimum inhibitory concentration of extracted secondary metabolites ranged from 8.000 mg/mL to 0.0625 mg/mL. Several samples showed inhibition at a concentration of 0.125 mg/mL, showing potential for development into compounds with promising bioactivities against pathogenic microorganisms, as seen in Table 2.

### 3.3. Effects of Bacterial Endophytes and Plant Parts on Cancer Cells

The extracts showed no notable anticancer activity against the glioblastoma and lung carcinoma cells as observed in Figure 2 and Figure 3. An increase in cell viability was observed at the highest concentrations, which is 100 µg/mL for each of the analyzed extracts.

## 4. Discussion

Endophytes are known to vary in diversity based on seasonal collection or sampling time, plant age, plant tissue type, and environment [21]. Recent studies have equally documented *Pseudomonas, Bacillus, Pantoea* and *Enterobacter* as bacterial endophytes in plants [31,32,33].

The assignment of bacterial isolates to a given species requires further phenotypic and molecular characterization. Based on this study, the isolates had 99% similarity to *Pantoea vagans* strain UFLA WCF767, *Pantoea agglomerans* strain FC2948, *Pantoea ananatis* strain P5, and *Pantoea eucalypti* strains EV2 and EV4. There are reports showing the association of *Pantoea* species with plants [34]. However, *Pantoea agglomerans* has been shown to act as a biocontrol agent on cotton [35], produce antibiotics that inhibit *Erwinia amylovora* [36], and promote plant growth [37,38].

Metabolites bearing antibiotic activity can be defined as low-molecular-weight organic natural substances, which are characterized by a molecular weight below 900 Dalton, made by microorganisms that are active at low concentrations against other microorganisms [13]. For example, Gentamicin, which is produced by the fermentation of *Micromonospora purpurea*, is known to be active at concentration 0.25 mg/L against *E. coli* [39]. Endophytes are believed to carry out a resistance mechanism to overcome pathogenic invasion by producing secondary metabolites [40]. *P. ananatis* extracts, when compared to other endophytes in this study, had the most significant MIC values between 0.0625 and 4.000 mg/mL. The most significant MIC value of 0.0625 mg/mL was seen on pathogenic *Staphylococcus aureus*. Other notable inhibitory activities were seen on *B. cereus, M. marinum, M. smegmatis,* and *S. epidermidis* with a MIC value of 0.125 mg/mL and *B. subtilis* with a MIC of 0.250 mg/mL. Antimicrobial activity of a bioactive compound from *P. ananatis* has been reported in a previous study by [41], which could support the notable inhibitory activity of its crude secondary metabolites in this study.

*P. vagans* extracts had MIC values ranging from 0.125 to 8.000 mg/mL. The most susceptible microbes were *B. subtilis, M. smegmatis*, and *S. aureus*, with a MIC value of 0.125 mg/mL. A notable MIC value of 0.500 mg/mL was also observed for *B. cereus* and *S. epidermidis*. A peptide present in *P. vagans* has been reported to possess antibacterial activity [42] and this could serve as a justification for the antibacterial activity seen in the crude secondary metabolite extracts of *P. vagans*.

*P. eucalypti* extracts showed MIC values between 0.125 and 4.000 mg/mL. Noteworthy inhibition was observed for *M. smegmatis, S. aureus* and *S. epidermidis*, with a MIC value of 0.125 mg/mL. Other significant inhibitions were seen on *B. cereus, B. subtilis, E. coli*, and *M. marinum*, with a MIC value of 0.500 mg/mL. This, to the best of our knowledge, is an initial report on the antibacterial activity of secondary metabolites from *P. eucalypti*.

The stem of *S. mauritianum* which was the host of the isolated endophytes was also analyzed for antibacterial activity. The MIC values seen for the stem crude extracts ranged from 0.0625 to 2.000 mg/mL. The crude extract showed very significant activity against *S. aureus* with a MIC value of 0.0625 mg/mL, which is also the same MIC seen on *S. aureus* when *P. ananatis* secondary metabolites were tested against it. There was activity seen for *B. cereus, B. subtilis, M. smegmatis, P. aeruginosa* and *S. epidermidis*, with MICs of 0.125 mg/mL. A MIC of 0.500 mg/mL was seen for *E. coli, K. pneumonia, M. marinum*, and *P. vulgaris*. Several *Solanum* species have been previously reported to have antibacterial activities against *S. aureus, E. coli, P. aeruginosa, B. subtilis*, and other common bacterial pathogens [43,44,45,46,47]. These reports validate the results on the antibacterial activity of the stem of *S. mauritianum* reported in this study.

Crude extracts exhibiting activity at concentration 1 mg/mL or lower are considered active [48]. In this study, crude secondary metabolites from both the plant part and isolated endophytes showed notable antibacterial activities against pathogenic microbes. Antimicrobial activity of compounds produced by plant endophytes have previously been reported [49,50]. It has also been reported that endophytes are the chemical synthesizers within plants. Many of them are capable of synthesizing bioactive compounds that can be used by plants for defense against pathogens and some of these compounds have been reported to be useful for drug discovery [13,14]. This supports the inhibitory activities seen in both the plant part analyzed and the isolated endophytes. It can also be seen from the results that the Gram-positive bacteria were more susceptible to both plant and endophyte crude extracts than the Gram-negative bacteria. This could be due to the presence of the outer membrane in the Gram-negative organisms, which excludes certain drugs and antibiotics from penetrating the cell. This in part accounts for why Gram-negative bacteria are generally more resistant to antibiotics than the Gram-positive bacteria [51]. The results from this study suggest that the secondary metabolites from the *S. mauritianum* plant and its bacterial endophytes could be explored further for the development of new pharmaceutical products against pathogens.

Although it has been previously indicated that different plant parts in some other species of *Solanum* possess anticancer properties (*S. nigrum* leaves possess anticancer properties against Ehrlich ascites carcinoma cell (EACC) line and Hepatoma cell (HepG2) line [52], tomatidine and solasodine from *S. aculeastrum* were shown to have an inhibitory effect on HT-29 (colonic adenocarcinoma), HeLa (cervical carcinoma), and MCF-7 (breast adenocarcinoma) cells [53]), the stem crude extract of *S. mauritianum* showed no antiproliferative effect on both cancer cell lines used in this study. To the best of our knowledge, there is no previous report on the anticancer properties of any of the *Pantoea* endophytes reported in this study, although exopolysaccharides isolated from an endophyte, *Bacillus amyloliquefaciens*, showed anticancer activity against gastric carcinoma cell lines [54]. Other studies on anticancer properties of fungal endophytes have also been previously reported elsewhere [55,56,57]. The anticancer analysis of crude secondary metabolites from both endophytes and the plant part in this study showed no notable anticancer activity against the two cancer cell lines treated. It is reported that Glioblastoma (GB) and lung cancer are among the most lethal human cancers. GB tumor cells as well as lung cancer cells have been shown to exhibit drug resistance and are highly infiltrative [58,59,60]. This resistance could account for the results observed in this study. Resistance to treatment and poor survival have been attributed to the presence of cancer stem cells (CSCs) in GB [61,62] and in lung carcinoma cells [63].

## 5. Conclusions

Microbe–plant interactions are far from being fully understood. Nevertheless, more evidence shows plant-associated microorganisms provide substantial benefits to agriculture, industry, and the environment. In brief, this study determined the existence of a plant–microbe relationship in the stems of *S. mauritianum* and the antimicrobial activity of the crude extracts of the plant and microbes, which showed notable inhibitory activities. The antibacterial results show the potential use of endophytic bacteria for the isolation of pure bioactive compounds and possible drug discovery. Further research needs to be carried out to isolate and identify pure active compounds produced by endophytes.

## Figures and Tables

**Figure 1 ijerph-16-00602-f001:**
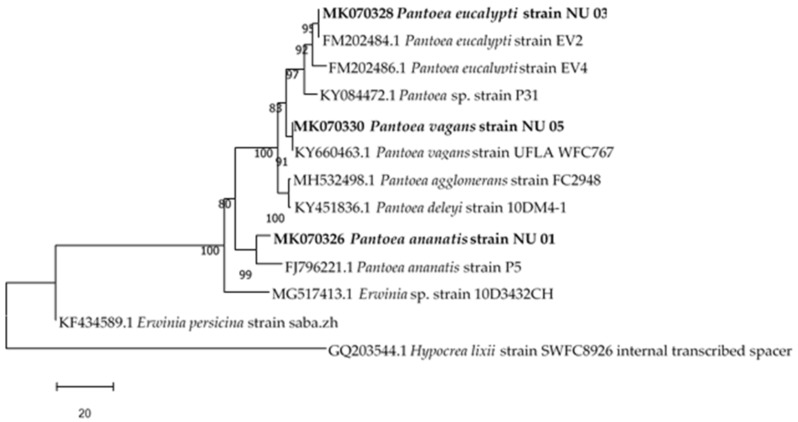
Neighbour-joining tree based on *16S* rRNA gene sequence of three endophytic bacteria, isolated from *S. mauritianum* and other similar species selected from GenBank.

**Figure 2 ijerph-16-00602-f002:**
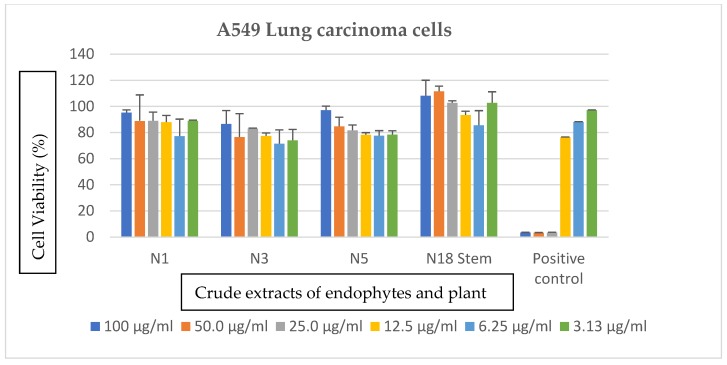
Cytotoxic effects of crude secondary metabolites from the stem of *S. mauritianum* and its bacterial endophytes on A549 Lung carcinoma cells tested at different concentrations, ranging from 100–3.13 µg/mL. Auranofin was used as positive control. N1 = *P. ananatis*, N3 = *P. eucalypti*, N5 = *P. vagans*, and N18 = *S. mauritianum* stem.

**Figure 3 ijerph-16-00602-f003:**
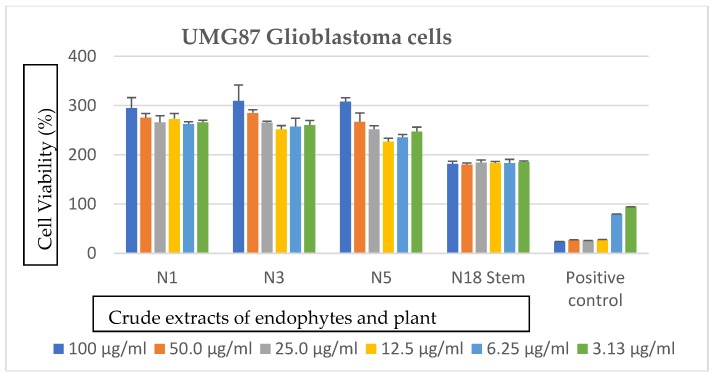
Cytotoxic effects of crude secondary metabolites from the stem of *S. mauritianum* and its bacterial endophytes on UMG87 glioblastoma cells tested at different concentrations ranging from 100–3.13 µg/mL. Auranofin was used as positive control. N1 = *P. ananatis*, N3 = *P. eucalypti*, N5 = *P. vagans*, and N18 = *S. mauritianum* stem.

**Table 1 ijerph-16-00602-t001:** Identification of bacterial endophytes from the stems of *Solanum mauritianum.*

Bacterial Isolate	Accession Number	Closest Relatives in the National Center for Biotechnology Information (NCBI)	Percentage Similarity	Classification	Macroscopic Identification	Microscopic Identification
NU 01	MK070326	*Pantoea ananatis* strain P5 (FJ796221)	99	*Pantoea ananatis*	Round, white, raised and entire colonies	Gram negative, rods
NU 03	MK070328	*Pantoea eucalypti* strain EV2 (FM202484)	99	*Pantoea eucalypti*	Round, Creamy, irregular and raised colonies	Gram negative, rods
NU 05	MK070330	*Pantoea vagans* strain UFLA WFC767 (KY660463)	99	*Pantoea vagans*	Round, translucent, convex and entire colonies	Gram negative, rods

**Table 2 ijerph-16-00602-t002:** Minimum Inhibitory Concentration (MIC) values of the antibacterial activity test carried out on the secondary metabolites, extracted from the identified endophytes and the plant part.

Test Organism	Gram Reaction of Test Organisms	*P. vagans* MIC (mg/mL)	*P. ananatis* MIC (mg/mL)	*P. eucalypti* MIC (mg/mL)	Plant Part (Stem) MIC (mg/mL)	Positive Control (Streptomycin) MIC (µg/mL)
*B. cereus*	Positive	0.5000	0.1250	0.5000	0.1250	0.03125
*B. subtilis*	Positive	0.1250	0.2500	0.5000	0.1250	0.03125
*E. aerogenes*	Negative	8.0000	4.0000	4.0000	2.0000	0.12500
*E. coli*	Negative	1.0000	0.5000	0.5000	0.5000	0.12500
*K. pneumoniae*	Negative	8.0000	2.0000	4.0000	0.5000	0.12500
*M. marinum*	Positive	0.5000	0.1250	0.5000	0.5000	0.06250
*M. smegmatis*	Positive	0.1250	0.1250	0.1250	0.1250	0.06250
*P. vulgaris*	Negative	4.0000	0.5000	4.0000	0.5000	0.12500
*P. aeruginosa*	Negative	4.0000	1.0000	2.0000	0.1250	0.12500
*S. aureus*	Positive	0.1250	0.0625	0.1250	0.0625	0.03125
*S. epidermidis*	Positive	0.5000	0.1250	0.1250	0.1250	0.06250

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
