# Peer review of "Antibacterial Activities of Crude Secondary Metabolite Extracts from Pantoea Species Obtained from the Stem of Solanum mauritianum and Their Effects on Two Cancer Cell Lines"

_ijerph, 2019, doi:10.3390/ijerph16040602_

Round 1

Reviewer 1 Report

In the manuscript entitled “Antibacterial activities of crude secondary metabolite isolated from Pantoea species obtained from the stem of Solanum mauritianum and their effects on two cancer cell lines” Nkemdinma Uche-Okereafor and co-workers have evaluated the anti-bacterial and anti-cancer activities of secondary metabolite crude extracts from the medicinal plant Solanum mauritianum and its bacterial endophytes. In the antibacterial assay, crude secondary metabolites of the endophytes and stem of S. mauritianum showed moderate antibacterial activities against  E. coli, S. aureus; K. pneumonia, and P. aeruginosa starins of bacteria. In the anti-cancer assay all compounds were completely inactive against A549 and UMG87 cell lines. Overall, the manuscript is of modest interest and could be published in IJERPH after some revision.

Queries:

1)      The authors should mention in table 2 that what are they measuring? Please add “MIC” in first row of table 2 where concentration (mg/mL) is written.

2)      The authors should mention in the manuscript why they have used “Auranofin” as standard drug in the anti-cancer assay instead of any known FDA approved anti-cancer drug. Auranofin is mainly used to treat rheumatoid arthritis although it has shown some anti-cancer affects against ovarian cancer cells.

Author Response

Kindly find my revisions below:

1. The authors should mention in table 2 that what are they measuring? Please add “MIC” in first row of table 2 where concentration (mg/mL) is written.

Answer: Addressed in table 2, line 233 and highlighted in red.

2. The authors should mention in the manuscript why they have used “Auranofin” as standard drug in the anti-cancer assay instead of any known FDA approved anti-cancer drug. Auranofin is mainly used to treat rheumatoid arthritis although it has shown some anti-cancer affects against ovarian cancer cells.

Answer: Auranofin has the potential to be repurposed for a lot of diseases. Auranofin has a high affinity for thiols and selenoproteins and is able to inhibit enzymes that reduce intracellular ROS such as TrxR, which have been shown to be an effective therapeutic target for cancer, parasitic infections and HIV. New drug discovery is an expensive and risky endeavor. It makes sense to look to old drugs that have already passed clinical trials and received FDA approval for other diseases.

These 2015 studies: “Auranofin-mediated inhibition of PI3K/AKT/mTOR axis and anticancer activity in non-small cell lung cancer cells” and “Auranofin: Repurposing an Old Drug for a Golden New Age” by Li et al., and Roder & Thomson respectively show that auranofin has anticancer activities. Therefore, auranofin was used as a control.

Reviewer 2 Report

Antibacterial activities of crude secondary metabolite isolated from Pantoea species obtained from the stem  of Solanum mauritianum and their effects on two cancer cell lines

Nkemdinma Uche-Okereafor, Tendani Sebola , Kudzanai Tapfuma , Lukhanyo Mekuto Ezekiel Green , Vuyo Mavumengwana

This is an interesting paper describing the isolation and the characterisation of bacterial endophytes from the plant solnum mauritianum. Those bacteria were cultured and their respective extracts together with the crude extract of the stem of the plant, were studied for their antibacterial and anticancer activities.

In general the paper is well written. However in order to enhance the quality of the paper I recommend that the authors to address the following:

In the introduction the authors referred to bioactive compounds from endophytes. It is important that the authors expand on the family of compounds (in terms of their chemical entity) that form those bioactive compounds

Furthermore the authors can also include bioactive compounds sourced from other medicinal plants where endophytes are present

Although the Material and Methods section is reasonably well written, there are some important details missing. In section 2.3.1 Isolation of secondary metabolites from endophytes: line 125, please insert the volume of acetone used to wash the resin. Was the acetone fraction evaporated to dryness? What was the weight and colour of the residue

The next paragraph (lines 127-130) is not clear at all. Was the extract purified by an ethylacetate/water extraction? 1:1 ratio (v/v) Is this ethylacetate/water?

Section 2.3.3 the stock solution (32 mg/mL) of the extract was prepared with DMSO (0.1%). Was culture media used for the serial dilution?

Resazurin was used for the antibacterial activity determination: Line 162 If the colour changes were visually inspected, how were the MIC values shown in table 2 obtained? Surely a spectrophotometer must have been used! A wavelength is required.

Section 2.3.4 Anti-cancer assays. Why were U87MG and A549 cell lines in this study? Where were they sourced? What concentration of the stock solution was prepared and how was the serial dilution carried out? With media or solvent? What was the rationale of using auranofin as a positive control? The latter is not usually used an anti-cancer agent!

Result section. Line 199- please insert the actual highest concentrations.

In the Table 2 title, please say the values are the MIC values. Has there been any statistical analysis carried out with these data?

Discussion Section. Line 240-242. It would be useful if the authors can actually insert values of low concentration of those active natural substances. Again the nature of the low molecular weight organic natural substances can be further highlighted. The same comment applies to ‘secondary metabolites’ through this section.

Author Response

Kindly find the corrections effected below:

1. In the introduction the authors referred to bioactive compounds from endophytes. It is important that the authors expand on the family of compounds (in terms of their chemical entity) that form those bioactive compounds

Response 1: This has been addressed in the article: lines 58 – 59 and highlighted in red.

2. Furthermore, the authors can also include bioactive compounds sourced from other medicinal plants where endophytes are present.

Response 2: This has been addressed in lines 62 – 65 and highlighted in red.

3. Although the Material and Methods section is reasonably well written, there are some important details missing. In section 2.3.1 Isolation of secondary metabolites from endophytes: line 125, please insert the volume of acetone used to wash the resin. Was the acetone fraction evaporated to dryness? What was the weight and colour of the residue?

The next paragraph (lines 127-130) is not clear at all. Was the extract purified by an ethyl acetate/water extraction? 1:1 ratio (v/v) Is this ethyl acetate/water?

Response 3: This has been addressed in lines 131 – 140 and highlighted in red.

4. Section 2.3.3 the stock solution (32 mg/mL) of the extract was prepared with DMSO (0.1%). Was culture media used for the serial dilution?

Response 4: Yes, culture media was used for dilution. This has been addressed in the article as seen on lines 167 – 168 and highlighted in red.

5. Resazurin was used for the antibacterial activity determination: Line 162 If the colour changes were visually inspected, how were the MIC values shown in table 2 obtained? Surely a spectrophotometer must have been used! A wavelength is required.

Response 5: MIC value recorded is defined as the lowest concentration of the assayed antimicrobial agent that inhibits the visible growth of the microorganism tested.

The MIC is the lowest concentration of antimicrobial agent that completely inhibits growth of the organism in tubes or micro- dilution wells as detected by the unaided eye (CLSI, Methods for Dilution Antimicrobial Susceptibility Tests for Bacteria that Grow Aerobically, Approved Standard, 9thed., CLSI document M07-A9. Clinical and Laboratory Standards Institute, 950 West Valley Road, Suite 2500, Wayne, Pennsylvania19087, USA,2012).

The plates were prepared in triplicate and incubated at 37 ˚C for 24 h. The dye was added and incubated for 2 hours and observed. The color change was then assessed visually. The growth was indicated by color changes from purple to pink where pink indicates bacterial growth and purple indicates inhibition. The lowest concentration after which color change occurred was taken as the MIC value. This is addressed in lines 173 – 174.

6. Section 2.3.4 Anti-cancer assays. Why were U87MG and A549 cell lines in this study? Where were they sourced? What concentration of the stock solution was prepared and how was the serial dilution carried out? With media or solvent? What was the rationale of using auranofin as a positive control? The latter is not usually used an anti-cancer agent!

a. Why were U87MG and A549 cell lines in this study?

Response 6a: They are known to be resistant and problematic therefore we decided to screen our extracts against them. Kindly see lines 308 – 311 in the manuscript.

b. Where were they sourced?

Response 6b: ATCC, addressed in line 176

c. What concentration of the stock solution was prepared and how was the serial dilution carried out? With media or solvent?

Response 6c: Stock solution concentration was 200 µg/mL. Serial dilution was done with media. This has been included in the article in lines 178 -179 and highlighted in red.

d. What was the rationale of using auranofin as a positive control?

Response 6d: The choice of auranofin as a control in this study is based on these 2015 studies “Auranofin-mediated inhibition of PI3K/AKT/mTOR axis and anticancer activity in non-small cell lung cancer cells” and “Auranofin: Repurposing an Old Drug for a Golden New Age” by Li et al., and Roder & Thomson respectively show that auranofin has significant anticancer properties.

7. Result section. Line 199- please insert the actual highest concentrations.

Response 7: Addressed in line 215.

8. In the Table 2 title, please say the values are the MIC values. Has there been any statistical analysis carried out with these data?

Response 8: Statistical analysis was not done. The MIC values show the lowest concentration at which the extracts were active against the test microbes.

9. Discussion Section. Line 240-242. It would be useful if the authors can actually insert values of low concentration of those active natural substances. Again, the nature of the low molecular weight organic natural substances can be further highlighted. The same comment applies to ‘secondary metabolites’ through this section.

Response 9: Has been addressed in line 256 - 259.
